# DIA-TTS: Deep-Inherited Attention-Based Text-to-Speech Synthesizer

**DOI:** 10.3390/e25010041

**Published:** 2022-12-26

**Authors:** Junxiao Yu, Zhengyuan Xu, Xu He, Jian Wang, Bin Liu, Rui Feng, Songsheng Zhu, Wei Wang, Jianqing Li

**Affiliations:** 1Jiangsu Province Engineering Research Center of Smart Wearable and Rehabilitation Devices, School of Biomedical Engineering and Informatics, Nanjing Medical University, Nanjing 211166, China; 2Department of Medical Engineering, Wannan Medical College, Wuhu 241002, China

**Keywords:** natural language processing, text-to-speech, deep learning, information theory, deep neural network, local-sensitive attention

## Abstract

Text-to-speech (TTS) synthesizers have been widely used as a vital assistive tool in various fields. Traditional sequence-to-sequence (seq2seq) TTS such as Tacotron2 uses a single soft attention mechanism for encoder and decoder alignment tasks, which is the biggest shortcoming that incorrectly or repeatedly generates words when dealing with long sentences. It may also generate sentences with run-on and wrong breaks regardless of punctuation marks, which causes the synthesized waveform to lack emotion and sound unnatural. In this paper, we propose an end-to-end neural generative TTS model that is based on the deep-inherited attention (DIA) mechanism along with an adjustable local-sensitive factor (LSF). The inheritance mechanism allows multiple iterations of the DIA by sharing the same training parameter, which tightens the token–frame correlation, as well as fastens the alignment process. In addition, LSF is adopted to enhance the context connection by expanding the DIA concentration region. In addition, a multi-RNN block is used in the decoder for better acoustic feature extraction and generation. Hidden-state information driven from the multi-RNN layers is utilized for attention alignment. The collaborative work of the DIA and multi-RNN layers contributes to outperformance in the high-quality prediction of the phrase breaks of the synthesized speech. We used WaveGlow as a vocoder for real-time, human-like audio synthesis. Human subjective experiments show that the DIA-TTS achieved a mean opinion score (MOS) of 4.48 in terms of naturalness. Ablation studies further prove the superiority of the DIA mechanism for the enhancement of phrase breaks and attention robustness.

## 1. Introduction

The text-to-speech (TTS) technique has played a vital role in interaction platforms to help generate natural and human-like speech based on input texts, especially for people with vision impairment [1,2,3]. It can serve as an assistive tool in many areas such as language education, voice navigation, clinical rehabilitation, and recreation [4,5]. Therefore, TTS has been one of the most attractive research topics among affective computing, natural language processing, and speech signal analysis. As a typical TTS method, the traditional hidden Markov Model (HMM)-based statistical parametric speech synthesis (SPSS) network [6] is a pipeline structure that involves many modules, each responsible for different tasks and trained individually, resulting in a combination of errors caused by each component. The complexity of the network raises training difficulties and elongates the inference time. Thanks to the fast development of deep learning and entropy theory, the modern TTS model has been simplified and greatly improved in terms of speech quality. Large numbers of computer engineers and researchers put their efforts in the direction of building advanced TTS mechanisms based on deep neural networks (DNNs). Traditional DNN-based TTS models, such as Char2Wav [7], Tacotron [8], Tacotron2 [9], Fast-Speech [10,11], DeepVoice [12,13,14], and so on, have achieved great success in generating human-like speech.

WaveNet [15] is an autoregression generative model based on the Pixel Convolutional Neural Network (PixelCNN) [16]. It introduces a dilated causal convolutional structure to extract acoustic latent features among long time intervals. Even though WaveNet has enormous potential in various TTS applications, the generation period is too long due to its complex model structure and autoregression property. To solve this deadly problem, Paine et al. [17] came up with a fast WaveNet that significantly reduces computational complexity by removing redundant convolution operations. However, WaveNet remains a typical backend neural network that depends on linguistic feature conditions from an existing fronted TTS. To further simplify the model construction and increase the generation efficiency, SampleRNN [18] was introduced by Mehri et al. and recognized as the first approach to the end-to-end TTS that unconditionally generates speech based on frame-level training. Following SampleRNN, Sotelo et. al [7] came up with Chr2Wav, an end-to-end TTS that contains a reader and a neural vocoder, where the vocoder is referred to as an extension of SampleRNN. Unlike traditional models, Char2Wav can synthesize speech directly from text without manual regulation. However, it is not identified as an end-to-end structure in the strict sense as the two components are trained separately. Not until the existence of Tacotron [8] was the train-from-scratch end-to-end model published.

Tacotron is a typical end-to-end model based on a DNN. It consists of three main blocks: an encoder for robust sequential representations of text; an attention-based decoder for text–audio alignment; and a vocoder to reconstruct the waveform audio based on the generated Mel-spectrogram. Text and audio are aligned through purely content-based attention [19] that lacks locational information, easily resulting in alignment errors. Tacotron2 is a modification of Tacotron which significantly simplifies the model complexity, while improving the performance of speech synthesis by replacing the post-net structure with an optimized WaveNet and enhancing the attention alignment by using local-sensitive attention [20]. Although Taocotron2 outperformed other models in synthesizing natural speech, unpredictable errors such as skipping, repeating, and mispronouncing words, and over and under generating prosody phrase breaks have commonly appeared. The most probable reason causing these shortcomings is the attention alignment problem [21,22,23,24]. Due to its autoregression generation scheme, Tacotron2 can easily lead to alignment errors or collapse when using a single local-sensitive attention, especially when dealing with long sentences. Liu et al. [21] tried to reduce the alignment exposure bias by introducing a new teacher–student training algorithm by using a distillation loss function in addition to the feature loss function, which improves the speech synthesis for out-of-domain text. Elias et al. [25,26] came up with a non-autoregressive structure that requires no supervised duration signals during training, significantly improving the generation accuracy and speech quality. Okamoto et al. [27] presented a Tacotron-based acoustic model that used a phoneme alignment mechanism to avoid alignment errors. All these state-of-the-art TTS models put forward valid solutions to the remaining shortcomings of traditional TTS, showing advantages in generating high-fidelity speech. However, to our knowledge, none of the above specifically deals with phrase break prediction while enhancing speech quality.

In this study, we introduce a deep-inherited attention-based TTS (DIA-TTS) that is superior to the traditional TTS models in (1) high speech quality, including word pronunciation and accuracy; (2) appropriate phrase break position, especially in circumstances when no punctuation marks are used in long sentences; and (3) greater detail in acoustic features (pitch, accent, and stress). The proposed DIA-TTS is a typical encoder–decoder structure that is connected by an improved local-sensitive attention with an adjustable local-sensitive factor (LSF). Instead of directly using acoustic, we use Mel-spectrogram as the input during the model training stage due to convenience. Furthermore, Mel-spectrogram contains more informative acoustic features by transferring signals in the time domain into the frequency domain using Fast Furious Transform (FFT) followed by 80-channel Mel-scale filters. The deep inheritance mechanism enables multi-processing of token–frame pairs by the iterative DIA, which fastens the alignment process, as well as enhances the alignment robustness. LSF controls the attention concentration region, ensuring a steady alignment process and a tight context connection. A multi-RNN layer is adopted in the decoder for deeper acoustic feature extraction; dropout layers were added to avoid increments in feature entropy by using Bernoulli distribution. The predicted Mel-spectrograms from the decoder are reconstructed into waveform audio by WaveGlow [28]. The integration of the proposed DIA mechanism promises an intelligent and natural speech synthesis by appropriately breaking the long sentences at the correct position, as well as improving the acoustic quality. Furthermore, the controllable LSF enables a steady and smooth alignment process, which is superior to traditional TTS models as the single soft attention mechanism commonly causes attention collapse. Even though many expressive TTS models show reliable performance in speech expression by adding additional modules such as Global Style Token (GST) [29], the proposed DIA-TTS model achieves satisfactory outcomes in TTS tasks in both short and long sentences without additional modules, which lessens the unnecessary errors caused by a separate training procedure.

## 2. Model Architecture

The proposed DIA-TTS is composed of three main components: (1) an encoder for input text feature extraction and analysis; (2) a DIA-based decoder to generate corresponding acoustic Mel-spectrogram; and (3) a WaveGlow-based vocoder to reconstruct the audio signals with the predicted Mel-spectrogram. The general model architecture is shown in Figure 1.

### 2.1. Encoder

The encoder of the DIA-TTS takes English characters as the input and returns a fixed-dimensional word embedding as the output. The input characters are first coded as a 512-dimensional vector by one-hot embedding and then passed through a stack of three one-dimensional convolutional layers with 512 filters and a kernel size of 5, followed by a bidirectional Long Short-Term Memory (LSTM) layer with 256 hidden units per direction. Dropout layers were added to each convolutional layer with a 0.5 probability using Bernoulli distribution, which avoids the increment in feature entropy caused by deepening the convolutional layers. The LSTM layer helps to find out contextual dependencies along time series.

### 2.2. DIA-Based Decoder

The DIA mechanism acts like a bridge that connects the input textural information from the encoder and the corresponding Mel-spectrogram generated by the decoder. Herein, an improved local-sensitive attention with an adjustable LSF controls the range of weights involved for convolutional computation among time series. It takes three components as inputs: the hidden-state information from the attention RNN layers, the contextual memory from the encoder, and the convolutional features computed from a concatenation of weights among adjacent time steps based on LSF. During training, an inheritance mechanism is introduced, by which the DIA is utilized repetitively for tribal times; each repetition inherits the training parameters from the last DIA while updating the hidden-state information from the corresponding attention RNN layers. This inheritance mechanism deepens the alignment scheme, as well as avoids attention loss by maintaining stabilization during training. Moreover, additional attention RNN layers are added to the DIA-based decoder to enhance the locality feature correlations between the linguistic context and acoustic expression during alignments by sending extra hidden-state information, computed after a residual connection [30], to the DIA for a further alignment procedure. Meanwhile, a combination of the outputs from the attention context and the hidden states of the previous attention RNN layer is used as the input for the next attention RNN layer. The entire process repeats the tribal times for each decoder step until the final attention context is produced and combined with the decoder’s hidden state as the input for the linear projection of the Mel-spectrogram or the prediction of the token stop. The output frame then undergoes the post-net that consists of five one-dimensional convolutional layers, each with 512 channels and a kernel size of 5.

#### 2.2.1. Tribal-Inherited Local-Sensitive Attention

Local-sensitive attention is a hybrid attention that combines the content and location information for effectively utilizing the monotonicity and locality of alignment [20]. It acts like an expansion of the additive attention mechanism but uses communitive weights from the previous decoder time steps as an additional local-sensitive feature. The overall attention equations can be expressed as:(1)$${m_{j}}_{j=1}^{L}=Encoder\left({x_{j}}_{j=1}^{L}\right)$$
(2)$$h_{i}=Attention\left(h_{i-1},c_{i-1},y_{i-1}\right)$$
(3)$$e_{i,j}=v^{T}\tan h\left(W_{h_{i}}+V_{m_{j}}+U_{f_{i,j}}+b\right)$$
(4)$$a_{i,j}=softmax\left(e_{i,j}\right)=\frac{\exp\left(e_{i,j}\right)}{\sum_{j=1}^{L}\exp\left(e_{i,j}\right)}$$
(5)$$c_{i}=\underset{j=1}{\sum }a_{i,j}m_{j}$$
(6)$$y_{i}=Decoder\left(y_{i-1},c_{i},h_{i}\right)$$
where ${\left\{x_{j}\right\}}_{j=1}^{L}$ is the input sequence with length L, $y_{i}$ is the output results, $m_{j}$ and $h_{i}$ represent the encoder and decoder’s hidden states, respectively, and $e_{i,j}$ is the attention energy, also called the alignment at time step i and alignment position j, and takes three components as inputs followed by a tanh activation and the linear normalization. $W_{h_{i}}$ and $V_{m_{j}}$ denote the query and key in the attention mechanism, which are the attention hidden states and encoder memory matrix in this study. Different from context-based attention, local-sensitive attention drives an additional feature that is sensitive to locational information as the third input. The $U_{f_{i,j}}$ represents the location-sensitive products that are computed from convolutional features along with the linear normalization of the attention weights $a_{i-1,j}$ from the previous decoder steps. $c_{i}$ indicates the attention context that summarizes a linear integration of the contextual feature matrix, attention weight concatenation, and the hidden-state vector. The local-sensitive attention not only takes contextual information as an important component during alignment but adds additional information regarding position to avoid attention collapse.

The proposed DIA-TTS improves the connection between the input text and the output Mel-spectrogram by using an optimized DIA mechanism that reloads the local-sensitive attention tribal times for every decoder step. Each reloading process takes the updated hidden-state components from the corresponding attention RNN layers and generates a new attention context. The newly produced attention context carries information from the previous alignment and again is used as the input of the attention RNN for the next iteration. The repetition of the attention inherits the overall training parameters from itself, affording a smooth and stable alignment. Moreover, instead of using commutative weights only from previous time steps, we introduced LSF, which controls the total number of weights among time series that are used for the computation of local-sensitive features. The modified local-sensitive component $U_{f_{i,j}}$ with LSF equal to 4 is described as follows:(7)$$y_{i}=Decoder\left(y_{i-1},c_{i},h_{i}\right)$$
(8)$$a_{cat}=CATENATE\left({\left\{a_{i-1,j}\right\}}_{j=1}^{L},\;{\left\{a_{i-2,j}\right\}}_{j=1}^{L},\;{\left\{a_{i-3,j}\right\}}_{j=1}^{L},a_{cum}\right)\left(LSF=4\right)$$
(9)$$U_{f_{i,j}}=Conv1D\left(a_{cat}\right)$$
where $a_{cum}$ represents the cumulation of the attention weights from the previous steps, and $a_{cat}$ is the concatenation of the weights (LSF = 4) from the last step $a_{i-1,j}$, two steps earlier $a_{i-2,j}$, three steps earlier $a_{i-3,j}$ and the overall cumulation of previous steps. Then, the concatenation of weights is computed by a one-dimensional convolutional layer with a filter size of 32 and kernel size of 31, followed by the linear normalization process. The increment in the weight proposition from the adjacent time step enhances location awareness and is robust to long inputs. In addition, by increasing LSF in DIA, the mild and smooth weight gradient gap for each token leads to a stabilized alignment procedure and avoids attention collapse and exposure. The novel DIA mechanism can integrate the encoder’s hidden states in multiple perspectives, resulting in better context vector generation, therefore leading to a better prediction of phrase breaks. The detail of the inherited local-sensitive attention is illustrated in Figure 2.

#### 2.2.2. Four-Layer RNN Block

The decoder is a DIA-based autoregressive RNN mechanism. It consists of four unidirectional LSTM layers: three attention RNN layers and one decoder RNN layer, each with 1024 hidden units in our model that act like a Mel-channel feature extractor and provide the details of the time series dependency between each Mel-spectrogram frame, as well as the time steps for attention alignment. During training, the predicted Mel-frames from the last iteration were passed through a pre-net consisting of two linear normalization layers with 256 hidden ReLU units, followed by a dropout layer with a frequency rate of 0.5. Then, the processed frames were fed into the four-layer RNN block that produces three attention hidden states and one decoder hidden state. The three attention hidden states were used as attention queries for repetitive attention alignment at each decoder time step. A residual connection was adopted to integrate attention with the hidden states at each iteration, as shown in Figure 2. This helps to achieve tight connections among neighboring tokens, as well as increases time dependencies. During each process, the attention RNN layers take a combination of latent information from the previous RNN layer and the corresponding attention context as the input. The additional RNN layers increase the depth of the network, which contributes to better acoustic feature extraction. The multi-layer RNN structure also serves to adjust the attention alignment dynamically, which optimizes the consistency of the decoding procedure, therefore avoiding robustness errors and unexpected outcomes, including under and over generation, wrong sentence duration, and inappropriate prosody phrase breaks.

### 2.3. WaveGlow Vocoder Model

WaveNet [15] was implemented as a typical vocoder for the reconstruction of audio from the predicted Mel-spectrogram in many TTS models and achieved a great performance [8,9,23]. However, the reconstruction process is quite time-consuming due to its simple-level autoregressive property, especially when dealing with long sentences. In this paper, we select WaveGlow [28], a flow-based end-to-end neural network, as our vocoder for transferring the synthesized Mel-spectrogram into real human natural waveform audio. WaveGlow is a combination of WaveNet and Glow [31] and is advanced in providing fast, efficient, and high-fidelity speech based on Mel-spectrogram and audio pairs. It utilizes a single cost function during training to maximize the likelihood of training data, simplifying the entire network and stabilizing the training process. Moreover, compared to WaveNet, WaveGlow is proven to have a faster inference, which realizes real-time speech synthesis without the need for autoregression. In this paper, we used the pre-trained WaveGlow model posted by NVIDIA.

## 3. Results

### 3.1. WaveGlow Vocoder Model

The acoustic signal in the frequency domain provides more information. Compared to the linear scale signal, a signal converted to a Mel-scale spectrogram is easier to analyze and contains more compact representations [9]. For the purpose of a better acoustic feature extraction, all the audio clips were transformed into frames by a Hann windowing of a 50-millisecond frame length with a 12.5-millisecond step shift, followed by a 2040-point Fast Fourier Transform (FFT) and 80-channel Mel-filters. The processed Mel-spectrogram was finally used for model training. The audio pre-processing diagram is shown in Figure 3.

In this paper, we utilized the LJ Speech dataset to train the model. The LJ Speech dataset is a public-domain speech dataset consisting of 13,100 short audio clips from a single female speaker. All the speech is spoken in US English, with a total duration of 23.55 h and 225,715 words. All the speech clips come with the corresponding translations in normalized text.

The proposed model was trained with a batch size of 32 distributed across two NVIDIA GEForce RTX 3090 GPUs, where each has 24 GB of memory for high-quality model training. The CUDA version for all the experiments is 11.7, and the model programming utilizes python 3.7 with pytorch version 1.9.1 + cu111. We applied teacher forcing during the training process by feeding the model with the ground truth. We used the Adam optimizer [32] with $\beta_{1}$ = 0.9 and $\beta_{2}$ = 0.999, ε = 10^−8^, and a fixed learning rate of 10^−3^. Instead of using the cross-entropy loss function, our loss adopted the mean square error (MSE) during training for autoregression issues. The whole training process involved 500 epochs and 200 k steps in total. The training and validation process is illustrated in Figure 4.

Compared to Tacotron2, the proposed DIA model shows a faster and more stabilized convergence during the training and validation process, indicating that increasing the DIA iteration depth speeds up the alignment.

### 3.2. Mel-Spectrogram Comparison Experiment

For the purpose of comparison, we generated Mel-spectrograms using Tacotron2 and the proposed model based on the same representative input text: “*The overwhelming majority of people in this country know how to sift the wheat from the chaff in what they hear and what they read*”. As for generative models, there is often no ground truth for a subjective comparison [33]; the reference Mel-spectrogram generated from real human acoustic speech illustrated in Figure 5 is simply used for visual comparisons as we would like to see how well the models perform to give an accurate phrase break at the appropriate position.

Figure 5 illustrates the comparison of Mel-spectrograms with different sources. Compared to the reference audio, the DIA-TTS model accurately predicted the two pauses, while Tacotron2 failed to add prosody breaks in the generated Mel-spectrogram. In addition, comparing (A) and (B), when taking a closer observation of the Mel-spectrograms, it is obvious to see that the proposed model did a better job dealing with details, giving a precious and clear representation of the Mel-scale frames in the high-frequency domain, while the Mel-spectrogram predicted by Tacoton2 is blurred. In addition, the yellow box with a dashed line, illustrated as (C) and (D), represents the small details of the last frame of the Mel-spectrogram synthesized by the two models, respectively, showing that the DIA-TTS has a better performance in accent since the Mel-spectrogram of accent presents a curve shape like that from the reference audio, as the bigger the curve, the stronger the accent of the audio is.

### 3.3. Human Subjective Evaluation Experiments

For the subjective speech evaluation, both Mean Opinion Score (MOS) and Comparative Mean Opinion Score (CMOS) experiments were conducted. During the MOS experiments, for each model, 100 synthesized speech clips were rated by eight different raters based on the Absolute Category Rating scale. Raters were asked to score each speech carefully, ranging from 1 to 5 with a 0.5-point increment, in terms of speech naturalness and accent performance. All the MOS experiments were independent, which means each group of raters only evaluated speech from one model or the ground truth. All the raters are native US English speakers with good hearing ability.

For the CMOS experiments, 50 groups of synthesized speech generated by Tacotron2 and the DIA-TTS model, respectively, were rated by eight raters for comparison, each group with the same text reference as the input. The score ranges from −3 to 3, where a negative score means Tacotron2 is better, while the positive score means the DIA-TTS is better, and 0 means both models perform the same. All the raters were blind testing when listening to the speech. The results are shown in Table 1.

From Table 1 we can see that the DIA-TTS model with WaveGlow as a vocoder achieves a MOS of 4.48 ± 0.045, which is higher than Tacoton2 in terms of naturalness and accent. The MOS of the proposed model with WaveGlow (MOS = 4.48) as the speech reconstructor is slightly higher than that using WaveNet (MOS = 4.41), indicating that WaveGlow performs better in the audio reconstruction process. In addition, the inference speed of WaveGlow is much faster than WaveNet, which is in agreement with [28]. When compared to Tacotron2, the CMOS proves that our model has a better performance in synthesizing speech, which is +0.257 points. A histogram of the CMOS from eight raters scoring a total of 100 synthesized speech clips is shown in Figure 6. The raters claimed that the speech generated by the proposed model had a good pronunciation and gave the appropriate prosody phrase breaks when no punctuation marks appeared, while Tacotron2 occasionally under-generated or mis-generated phrase breaks or mispronounced words. For instance, Tacotron2 mispronounced the present-tense ‘read’ into past-tense ‘read’ when the input text used the present verbs. Furthermore, the proposed model appeared to generate more human-like speech in terms of accent and intonation for both short and long sentences. Even though Tacotron2 did an excellent job in synthesizing intelligent speech for short sentences, the speech based on long text inputs sounded slightly robotic with no emotion.

### 3.4. Ablation Study

To optimize the model structure with an appropriate attention inheritance depth and LSF, we carried out the ablation studies on six models with a different attention inheritance depth and LSF. The inheritance depth is defined as the number of attention RNN layers in the decoder. During each training step, attention updates its input variances, including the hidden-stage information from the attention RNN layers. While each RNN layer renews the hidden states for the attention mechanism, the attention increases its inheritance depth as it again inherits the training parameters. Since Tacotron2 has one attention RNN layer and uses a single local-sensitive attention with a weight concatenation of the previous weight and the cumulative weight, it is regarded as the model with one attention inheritance depth and LSF equal to two. The model details can be found in Table 2.

During the ablation studies, the same representative input text from the following LJ Speech recording were transferred into speech by six different models: “*Visited Mr. Fauntleroy, my application for books for him not have been attended, I had no prayer book to give him*”. Figure 7 represents the Mel-spectrogram predicted by models with a different attention inheritance depth and LSF. Based on the Mel-spectrograms, we can see that the models with multiple inheritance depths correctly predict the right pause position, while Tacotron2 misses the first pause, as shown in the red box. Furthermore, the comparison of the attention alignments tells us that the proposed DIA-TTS model (inheritance depth = 3, LSF = 4) and Model 4 (inheritance depth = 4, LSF = 5) have brighter and clearer encoder–decoder alignment lines, indicating a stronger attention robustness. As the increment in attention inheritance depth elongates the inference process when synthesizing the same length of sentence, Model 4 costs an average of 1.625 s to generate one speech. However, the proposed DIA-TTS model produces relatively the same quality of speech but with a faster generation time of 1.310 s. Therefore, the DIA-TTS with an attention inheritance depth of three and LSF of four is considered the optimal one for speech synthesis. The results of the MOS from different models and the average inference time for 50 randomly selected sentences can be found in Table 2.

## 4. Discussion

In this paper, we introduced a neural end-to-end TTS model based on the DIA mechanism involving a wide range of communitive weight concatenation controlled by LSF. Instead of using one attention RNN layer in the traditional TTS decoder, we implemented three attention RNN layers, each correlated with the DIA mechanism. The outputs of the attention RNN layers went into the same attention, respectively, for the alignment process for each decoder time step. In this case, the DIA reused the training parameters and inherited the hidden-state information from each RNN layer’s tribal times, resulting in a repeat and deep process for each token. The deep inheritance of the hidden state’s information also improves the linguistic feature extraction, as well as enhances the correlation between the context input and audio output, which contributes to an accurate prediction of the Mel-spectrogram. In addition, to tighten the feature connection between adjacent tokens, we added additional convolutional features controlled by LSF from extra weight concatenation among multiple time steps, capturing more locational information during alignment.

The results of the model comparison experiments show that the proposed model is superior in accurately predicting phrase breaks at the appropriate position simply based on the textual input without assistance from the ground truth, demonstrating that the proposed model is capable of mimicking the way people talk. Since there is no objective evaluation for generative models, visual comparisons are of vital importance. The Mel-spectrogram comparison indicates that our model is advanced in producing acoustic details, especially in the high-frequency domain. This advantage is perceptually expressed when reconstructing the Mel-spectrogram into waveform audio. Compared with Tacotron2 (MOS = 4.38 ± 0.052), the proposed model achieved a MOS of 4.48 ± 0.045, indicating that the proposed model has a better performance in synthesizing high-fidelity speech. The CMOS experiment achieved a score of positive 0.257, representing that the speech synthesized by our model is more pleasant than that generated from Taocton2 in aspects of the fluency, articulation, and naturalness of the audio. To acquire a high-quality speech signal in a real-time process, we utilized WaveGlow as the vocoder for audio reconstruction based on the predicted Mel-spectrograms. In this task, WaveGlow receives a MOS of 4.48 ± 0.045, while WaveNet receives 4.41 ± 0.047, representing that WaveGlow has a slightly better performance in the reconstruction of acoustic signals but with a faster inference time.

The results of the ablation study on investigating the optimal number of attention inheritance depths and LSF prove that a model with three inheritance depths and an LSF of four achieves the best performance by a comparison of the Mel-spectrogram, attention alignment, MOS, and the inference time. Among the six TTS models, Taocotron2, the proposed DIA model, and Model 4 produce a relatively clear and bright attention alignment line, representing good attention robustness and a tight connection between the encoder and decoder. Both the proposed DIA model and Model 4 receive a satisfactory MOS of 4.48 ± 0.045 and 4.49 ± 0.043, which are higher than the others. However, when taking training time duration as a factor into consideration, the proposed DIA-TTS is considered the optimal model for speech synthesis. Since the increment in RNN layers slows down the synthesis process, Li et al. [34] replaced the RNN model with a transformer. We intend to investigate a potential TTS structure with a different inheritance depth but a faster inference duration for future work.

In conclusion, we introduced a novel DIA-TTS synthesizer that can correctly predict phrase breaks and generate audio with high intelligibility and naturalness in real-time speech synthesis applications. The core invention of this work is based on a novel deep-inherited local-sensitive attention mechanism, which fastens the attention alignment process, enhances robustness, and ensures stabilization.

## Figures and Tables

**Figure 1 entropy-25-00041-f001:**
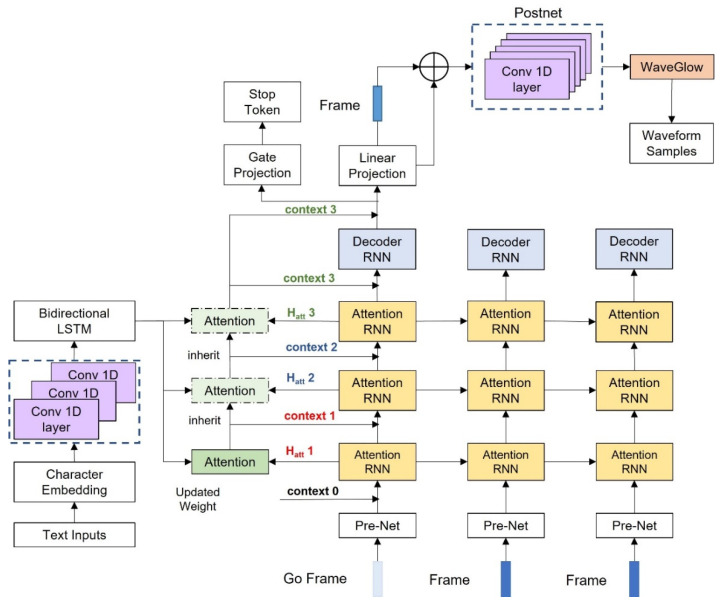
The general architecture of the DIA-TTS model, which consists of an encoder, a DIA-based decoder, and a WaveGlow-based vocoder.

**Figure 2 entropy-25-00041-f002:**
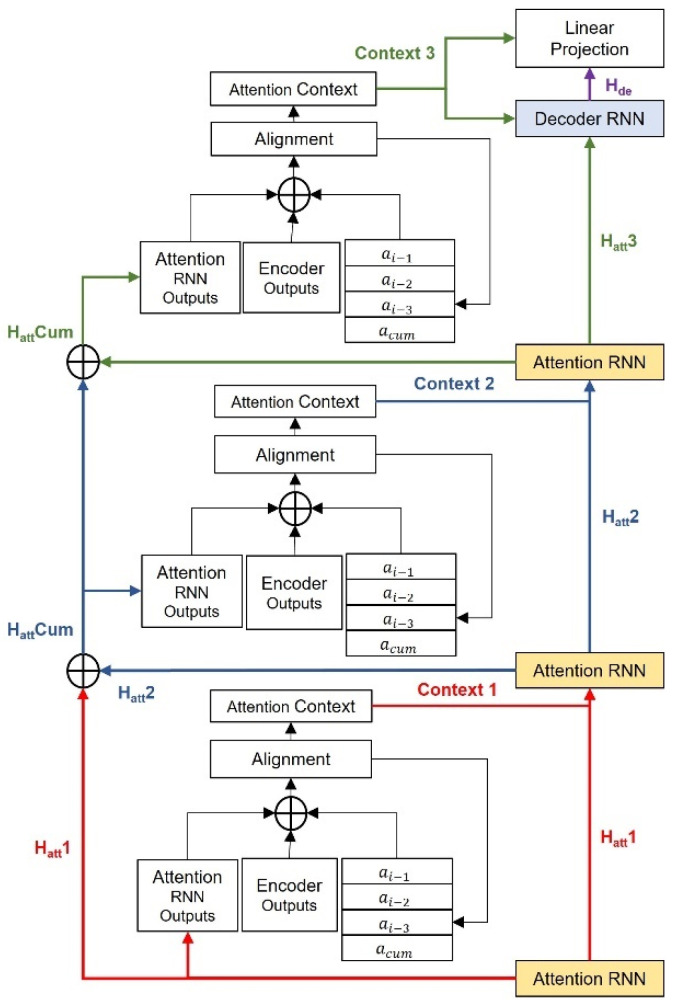
The architecture of the DIA mechanism. The improved local-sensitive attention takes three components as the input: the hidden state from the corresponding attention RNN, the content-based memory from the encoder output, and the convolutional features with an LSF of four.

**Figure 3 entropy-25-00041-f003:**
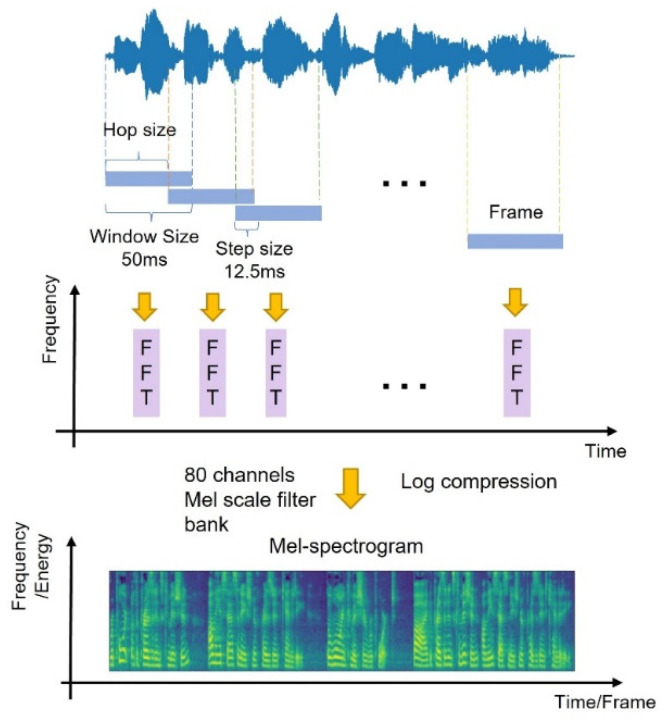
The flow diagram of audio pre-processing.

**Figure 4 entropy-25-00041-f004:**
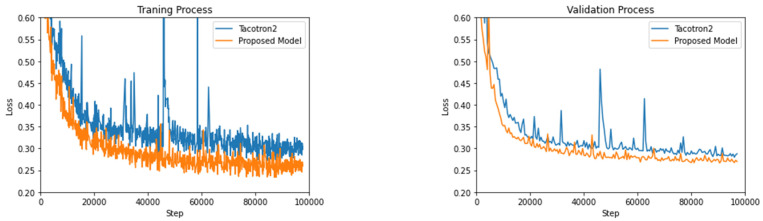
Convergence rate in loss using Tacotron2 (blue) and the proposed inherited Attention Tacotron2 (orange) during training and validation processes.

**Figure 5 entropy-25-00041-f005:**
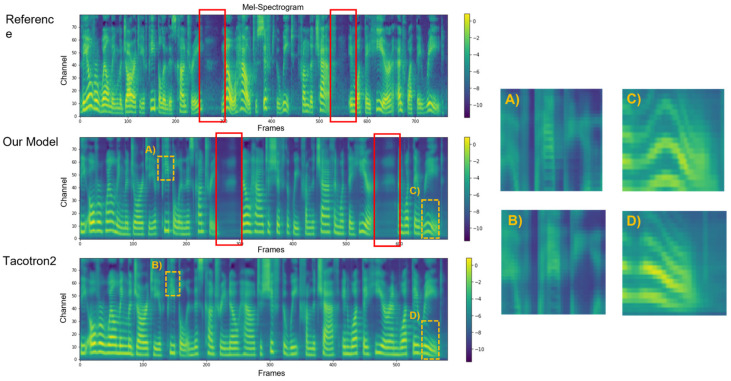
Mel-spectrograms from the reference audio, synthesized by the DIA-TTS model, and synthesized by Tacotron2. The red box represents the break occurrence, where two are obviously found in the reference and two in the Mel-spectrogram generated by the proposed model, yet none appears in the Mel-spectrogram synthesized by Tacotron2. The square pictures on the right represent details of the Mel-spectrogram. Compared to Tacotron2 (**B**), the proposed model generated more details (**A**) towards the high-frequency components. Furthermore, the DIA-TTS gave the beautiful, curved shape of each frame that corresponds to the accent when reconstructed into waveform audio (**C**) when compared to Tacotron2 (**D**).

**Figure 6 entropy-25-00041-f006:**
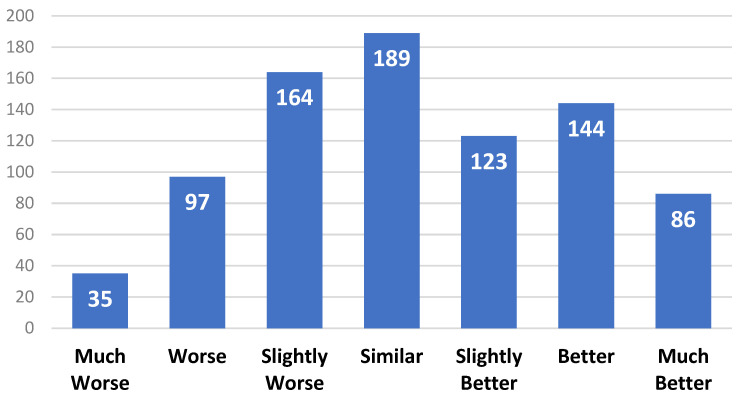
Histogram of CMOS results when comparing Tacotron2 and the DIA-TTS.

**Figure 7 entropy-25-00041-f007:**
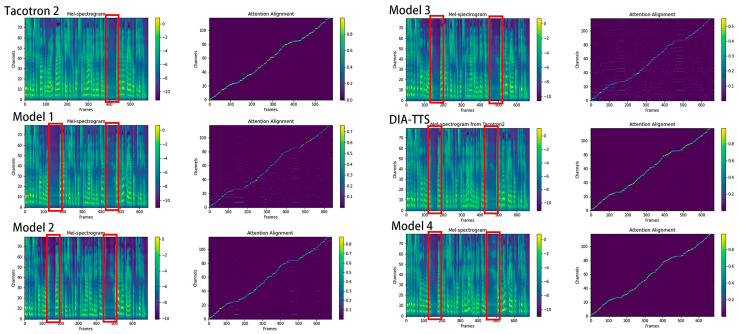
The ablation study of the proposed model in aspects of number of attention inheritance depth and LSF. For each model illustration, the left column represents the synthesized Mel-spectrogram, where the red boxes indicate the correct sentence break; the right column represents the corresponding attention alignments.

**Table 1 entropy-25-00041-t001:** The MOS evaluations with 95% confidence intervals.

System	MOS	CMOS
Tacotron2 + WaveGlow	4.38 ± 0.052	Reference
DIA-TTS + WaveGlow	**4.48 ± 0.045**	+0.257
DIA-TTS + WaveNet	4.41 ± 0.047	N/A
Ground Truth	4.51 ± 0.039	N/A

**Table 2 entropy-25-00041-t002:** MOS of synthesized speech for different TTS models with 95% confidence interval and the average inference time for 50 sentences.

Model	Inheritance Depth	LSF	MOS	Inference Time/s
Tacotron2	1	2	4.38 ± 0.052	0.724
Model 1	2	2	4.38 ± 0.027	0.966
Model 2	2	3	4.40 ± 0.052	1.032
Model 3	3	3	4.43 ± 0.044	1.270
DIA-TTS	3	4	4.48 ± 0.045	1.310
Model 4	4	5	4.49 ± 0.043	1.625

## Data Availability

Not applicable.

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
