# Peer review of "DIA-TTS: Deep-Inherited Attention-Based Text-to-Speech Synthesizer"

_entropy, 2022, doi:10.3390/e25010041_

Round 1

Reviewer 1 Report

This paper proposes a novel deep learning method called DIA-TTS model for the problem of text-to-speech (TTS). The major novelty of the proposed DIA-TTS is the deep-inherited attention (DIA) mechanism. DIA enables the DIA-TTS to learn the optimized local-sensitive attention parameters, which tightens the token-frame correlation as well as fastens the alignment process. Hence, DIA-TTS can converge quickly and has better correctness for generating long sentences. Experimental results showed that the proposed DIA-TTS achieved more promising performance than recent state-ot-the-art TTS methods, e.g., Tacotron2. In summary, the paper is well organized and written and easy to follow. However, some minor comments should be carefully addressed before acceptance.

(1) In Line 103, Page 3, is STFT the abbreviation of short-time Fourier Transform?

(2) As shown in Figure 1, the model has a postnet structure and a pre-net structure. However, I cannot find the details of both strucutres in the manuscript. Please provide the details in the revised version such that the readers can better understand the proposed method.

(3) Another concern is about the Attention RNN module in the proposed model. I found its output is denoted by Hatt, while denoted by Si in the manuscript. Please check the manuscript to avoid the abuse of variables.

(4) Several typo errors and improper expressions should be corrected. For example, in Line 272, Page 8, “issueres” -> “issues”; in Lines 63-64, Page 2, “…, Sotelo et.al came up with Chr2Wav [7] an end-to-end TTS that contains …” -> ”…, Sotelo et al. [7] came up with an end-to-end model called Chr2Wav containing …”. I strongly suggest the authors to carefully and seriously proofread the mansucript again to correct these mistakes.

Author Response

We thank the reviewer for the valuable comments and suggestions. Please find the attached file as our response to all the comments. The response to comments are marked in red.

Reviewer 2 Report

The authors propose a novel text-to-speech (TTS) synthesizer that interestingly adoptes a deep-inherited attention (DIA) mechanism for high-fidelity speech generation. The key part of the proposed model, i.e., DIA, provides a fast and steady convergence for the model training by enabling the proposed model to inherit previous training parameters and hidden state information. To evaluate the proposed method, the authors conducted extensive experiments, which demonstrated that the proposed method can learn a better correlation between context input and acoustic output resulting satisfactory speech signals, especially for long sentences. Consequently, I suggest the paper can be accepted. Before that, I still have several concerns.

My first concern is about the DIA mechanism. In DIA, the inheritance depth is defined as the number of RNN layers in the decoder. It would be better to provide more detailed description on this point in the revised manuscript.

Another concern is about the detail of implementation in the experiments. How to set parameters for the proposed method in the experiments? Which version of the CUDA? Pytorch or TensorFlow? Which Version? Providing these details are very important because readers can reimplement the experiments in the future.

Last but not least, the authors should check the manuscript again. Several grammar and typo errors exist. 1) Line 279, “deep inherited attention” should be presented as the abrrevation express “DIA”. 2) Line 376, “represent” should be written as “represents”. 3) In Table 1, “Group Truth” should be “Ground Truth”.

Author Response

We thank the reviewer for the valuable comments and suggestions. The response to the comments is marked in red. Please see the attachment. 
